# Rationale for Polyclonal Intravenous Immunoglobulin Adjunctive Therapy in COVID-19 Patients: Report of a Structured Multidisciplinary Consensus

**DOI:** 10.3390/jcm10163500

**Published:** 2021-08-08

**Authors:** Irene Coloretti, Giorgio Berlot, Stefano Busani, Francesco Giuseppe De Rosa, Abele Donati, Francesco Forfori, Giacomo Grasselli, Lucia Mirabella, Carlo Tascini, Pierluigi Viale, Massimo Girardis

**Affiliations:** 1Anaesthesia and Intensive Care Department, University Hospital of Modena, 41124 Modena, Italy; icoloretti@unimore.it (I.C.); stefano.busani@unimore.it (S.B.); 2Anestesia and Intensive Care Department, University Hospital of Trieste, 34127 Trieste, Italy; giorgio.berlot@asugi.sanita.fvg.it; 3Infectious Disease Department, Torino SCU, Hospital of Asti, 14100 Asti, Italy; francescogiuseppe.derosa@unito.it; 4Anaesthesia and Intensive Care Department, University Hospital of Ancona, 60127 Ancona, Italy; a.donati@univpm.it; 5Anaesthesia and Intensive Care Department, University Hospital of Pisa, 56124 Pisa, Italy; francesco.forfori@unipi.it; 6Anaesthesia and Intensive Care Department, University Hospital of Milan, 20122 Milano, Italy; giacomo.grasselli@unimi.it; 7Anaesthesia and Intensive Care Department, University Hospital of Foggia, 71122 Foggia, Italy; lucia.mirabella@unifg.it; 8Infectious Disease Department, University Hospital of Udine, 33100 Udine, Italy; c.tascini@gmail.com; 9Infectious Disease Department, University Hospital of Bologna, 40126 Bologna, Italy; pierluigi.viale@unibo.it

**Keywords:** respiratory failure, COVID-19, intravenous immunoglobulin therapy

## Abstract

Introduction: Adjunctive therapy with polyclonal intravenous immunoglobins (IVIg) is currently used for preventing or managing infections and sepsis, especially in immunocompromised patients. The pathobiology of COVID-19 and the mechanisms of action of Ig led to the consideration of this adjunctive therapy, including in patients with respiratory failure due to the SARS-CoV-2 infection. This manuscript reports the rationale, the available data and the results of a structured consensus on intravenous Ig therapy in patients with severe COVID-19. Methods: A panel of multidisciplinary experts defined the clinical phenotypes of COVID-19 patients with severe respiratory failure and, after literature review, voted for the agreement on the rationale and the potential role of IVIg therapy for each phenotype. Due to the scarce evidence available, a modified RAND/UCLA appropriateness method was used. Results: Three different phenotypes of COVID-19 patients with severe respiratory failure were identified: patients with an abrupt and dysregulated hyperinflammatory response (early phase), patients with suspected immune paralysis (late phase) and patients with sepsis due to a hospital-acquired superinfection (sepsis by bacterial superinfection). The rationale for intravenous Ig therapy in the early phase was considered uncertain whereas the panelists considered its use in the late phase and patients with sepsis/septic shock by bacterial superinfection appropriate. Conclusion: As with other immunotherapies, IVIg adjunctive therapy may have a potential role in the management of COVID-19 patients. The ongoing trials will clarify the appropriate target population and the true effectiveness.

## 1. Introduction

Since 20 February 2020, Italy has been overwhelmed by the SARS-CoV-2 virus outbreak, and several patients with interstitial pneumonia and respiratory failure requiring mechanical ventilation were admitted to intensive care units (ICUs), threatening the capability of healthcare systems to handle this amount of critical patients [1]. Unfortunately, so far, there are few validated therapies to prevent or treat the severe acute respiratory distress syndrome (ARDS) caused by this novel virus and thus the case fatality rate in patients admitted to ICU is extremely high [2,3,4,5,6,7]. Therefore, along with the maintenance of vital functions by supportive treatments, effective therapies for COVID-19 are urgently needed. 

In the previous months, the scientific community provided a tremendous improvemen and chemokines (the so-called cytokine storm) with a pivotal role in lung tissue damage, increase in vascular permeability and clots formation, akin to secondary hemophagocytic lymphohistiocytosis (sHLH) and macrophage activation syndrome (MAS) [8,9,10,11]. The COVID-19-associated cytokine storm is associated with elevated plasma levels of IL-6, IL-1 and TNF-α, as well as of ferritin and other inflammatory biomarkers. However, a recent study reporting cytokine levels in different subsets of critically ill patients showed that in COVID-19 patients with ARDS, the circulating levels of these cytokines were lower compared to those measured in patients with bacterial sepsis and similar to those with other causes of ARDS, trauma and out-of-hospital cardiac arrest [12]. Despite the limitations of the study, this may suggest that severe COVID-19 llness may be more than a cytokine storm, acting with more complex mechanisms involving innate and cellular immune response [13]. Different studies have explored the derangements of the immune system during COVID-19 and the associations with the outcome [14,15]. First, a key feature of severely ill patients with COVID-19 is represented by progressive lymphopenia with marked CD-4 and CD-8 T cell exhaustion [16,17,18]. 

More recently, COVID-19 clinical syndrome and related immunopathogenesis have been compared with sepsis, recalling the need to target the underlying and shared impairment of protective T cell immunity while suppressing the emergent cytokine storm [19,20,21,22]. Indeed, Hotchkiss et al. described the similarities between the course of immune activation and suppression during sepsis and COVID-19, suggesting that in the former, the hyperinflammatory peak may be higher, and the immunosuppressive phase may be deeper and earlier in the latter. This trend may be also reinforced by the use of immunosuppressive agents (e.g., steroids and cytokine-blocking agents) introduced in the treatment of patients with COVID-19 and respiratory failure [21]. Further investigations are warranted to clarify the relationships between these clinic and immunologic features in severe COVID-19 patients, possibly indicating the need to modulate the host immune response with immunotherapeutic treatments.

## 2. Adjunctive Immunoglobulin Therapy

As described above, sepsis and septic shock result from complex dysregulation of the inflammatory and immune response [22] that is quite similar to immunological derangement observed in critical COVID-19 patients. Immunoglobulins have pleiotropic effects on the inflammatory–immune response including toxin scavenging, microbial phagocytosis, anti-inflammatory effects and antiapoptotic actions on immune cells [21,23,24,25,26,27]. Although guidelines do not indicate the use of intravenous polyclonal immunoglobulin (IVIg) in patients with bacterial infections [28], several studies showed a potential benefit in patients with sepsis and septic shock [24,25,29,30,31], and IVIg are commonly used as adjunctive therapy in immunocompromised patients with infections [26,32]. Therefore, adjunctive therapy with IVIg may also have rationale in the management of COVID-19 patients that depends on the disease phase and the related pathobiological phenotype. For instance, although the role of persistent viremia and viral activity in tissues is unclear, Ig may have a role in the early phases of COVID-19 by reducing the viral burden and by scavenging or downregulating the production of high levels of inflammatory mediators. In the late phases, especially in ICU patients with secondary bacterial infections, IVIg may have an important synergic activity in the empowerment of antibiotic efficacy and in supporting the overt immune dysfunction [33] (Figure 1).

Although the multifaceted immunomodulant properties of IVIg could have beneficial effects in COVID-19 patients [27,34], clinical data supporting the use of adjunctive therapy with IVIg in these patients are few and limited only to the use of standard polyclonal IVIg [35,36,37,38,39,40,41]. A recent multicenter retrospective cohort study evaluated the efficacy of adjunctive therapy with IVIg by comparing 172 critically ill COVID-19 patients who received IVIg at the dose of 0.1–0.5 g/kg/day for 5–15 days with 151 critically ill COVID-19 patients who did not receive IVIg [35]. They observed that early administration (≤7 days post-admission) of high-dose IVIg improves the prognosis of critical-type patients with COVID-19 with a 20% absolute risk reduction in 28-day mortality. A small pilot randomized controlled study (16 vs. 17 patients) showed that 0.5g IVIg/kg daily for 3 days with concomitant methylprednisolone reduces the progression of respiratory failure requiring mechanical ventilation and improved oxygenation at 7 days in COVID-19 patients with PaO2/FiO_2_ < 140 [38]. A retrospective study also confirmed the therapeutic benefits of IVIg when therapy was initiated early [39]. On the other hand, a randomized controlled trial including 84 patients (52 treated patients vs. 32 controls) with severe COVID-19 pneumonia did not report any benefit of the use of IVIg at the dose of 0.4 g/kg/day for three days [41]. Indeed, the interpretation of the results reported in the abovementioned studies is problematic because of the high heterogeneity of study designs, with different doses and time of IVIg administration and the concomitant use of steroids. 

Noteworthily, the safety profile of IVIg preparations in COVID-19 patients seems to be high and similar to that observed in other patient populations where few adverse events are reported [42,43,44,45].

This manuscript reports the results of a structured process of consensus among experts aimed to discuss the rationale for adjunctive therapy with intravenous immunoglobulins and identify the phenotype of COVID-19 patients who could benefit the most based on the pathobiology of COVID-19 and pharmacological effects of adjunctive IVIg therapy.

## 3. Consensus Methodology

The moderator (M.G.) selected nine experts in the field of intensive care medicine and infectious diseases to create a multidisciplinary panel. All the panelists had a strong research profile with robust clinical experience in the management of COVID-19 patients and adjunctive IVIg therapy and were well-experienced in procedures of structured consensus.

In the first meeting, after an initial discussion of the main difficulties in COVID-19 management, the panelists defined the methods for the consensus and the different phenotypes of COVID-19 patients with severe respiratory failure based on the time course of the disease, clinical presentations and the underlying pathophysiological features. Three different clinical scenarios were identified: (i) early phase: patient with an abrupt and dysregulated hyperinflammatory response; (ii) late phase: patient with suspected immune dysfunction or immune paralysis; (iii) sepsis by bacterial superinfection: patient with sepsis or septic shock caused by hospital-acquired superinfection. For this consensus, the SARS-CoV-2 infection was defined as a positive result of a real-time reverse transcription polymerase chain reaction (RT-PCR) assay of nasopharyngeal swabs or lower respiratory tract specimens. Moderate-to-severe ARDS was defined as new or worsening respiratory failure with bilateral opacities and PaO_2_/FiO_2_ ≤ 200 mmHg with positive end expiratory pressure ≥ 5 cmH_2_O not fully explained by cardiac failure, fluid overload, pleural effusions and lobar or lung collapse [46].

The treatment selected was polyclonal intravenous immunoglobulin (IVIg) administration, including polyclonal IgG preparations that contain at least 96% of polyclonal IgG and IgM-enriched preparations where the composition is polyclonal IgG (76%), IgM 12%) and IgA (12%). IVIgM preparations, compared to IVIg, seem to provide a better clinical effect in septic patients [47,48] due to the IgM component and its fundamental role in innate immune response [49].

Due to the scarce evidence available on immunoglobulin treatment for COVID-19 patients with respiratory failure, the panelists decided to use a modified semiquantitative RAND/UCLA appropriateness method [50]. This semiquantitative method allows each part of the panel to express an opinion not influenced by other experts and supply the lack of evidence with the experience and opinion of the panelists.

The systematic review of literature according to population, treatment and the relevant outcome was performed using three electronic databases: PubMed, EMBASE and Scopus. All the literature material was readily available at any time for all the panelists. The coordinator of the panel (M.G.) analyzed and summarized the literature in the table of evidence that was available to all the panelists before the second meeting. In the second meeting, the summary of evidence was presented by the coordinator (M.G.) and discussed by all the panelists. During this meeting, the list of clinical scenarios and treatments was better redefined to avoid uncertainties in the rating procedures. An online voting system was used for the final anonymous vote. The panelists had to rate each clinical scenario as ‘appropriate’, ‘inappropriate’ or ‘uncertain’ on a scale from 1 to 9 points, with 1 = ‘completely inappropriate’ and 9 = ‘fully appropriate’. The median of the ratings of all the panelists was calculated, and we defined as inappropriate a scenario with the median value from 1 to 3, ‘uncertain’—from 4 to 6, ‘appropriate’—from 7 to 9. ‘Disagreement’ for each scenario was defined when more than three panelists gave ratings outside the 3-point region (1–3, 4–6 or 7–9) containing the median [50]. 

## 4. Consensus Results 

See Figure 2.

Scenario 1, early phase (see Figure 2): COVID-19 interstitial pneumonia with an abrupt and dysregulated hyperinflammatory response.

Description of the scenario: Patient admitted to hospital after <24 h from the onset of symptoms with rapid worsening of acute respiratory failure (PaO_2_/FiO_2_ < 200 mmHg) caused by SARS-CoV-2-related interstitial pneumonia and high plasma levels of such inflammatory parameters as C-reactive protein, ferritin and IL-6. 

Questions:(1)In COVID-19 patients with acute respiratory failure and severe hyperinflammatory response, how appropriate is the early (within 6–12 h) therapy with polyclonal intravenous immunoglobulins?Consensus rating: uncertain; median score, 6 (IQR, 5–7); disagreement: yes.(2)In COVID-19 patients with acute respiratory failure, severe hyperinflammatory response and decision to use polyclonal intravenous immunoglobulins, how appropriate is the use of preparations including also the IgM component?Consensus rating: appropriate; median score, 8 (IQR, 6–9); disagreement: no.

Rationale:

In the early phases of COVID-19, the proinflammatory response often predominates, with the massive production of proinflammatory cytokines such as tumor necrosis factor (TNF)-a, IL-8 and IL-6 that stimulate the effector functions of neutrophils, macrophages and Th1 cells. This dysregulated inflammatory response is similar to that observed in the early phase of sepsis, especially in the conditions leading to toxic shock syndrome, such as pneumococcal and meningococcal invasive disease or necrotizing fasciitis [51]. In this setting, a potential benefit of the early use of adjunctive therapy with IVIg, particularly of IgM-enriched preparations, has been shown in numerous clinical experiences [52,53,54], and its use is supported by many experts despite the lack of definitive evidence. As described above, the rationale for IVIg administration in patients with a high inflammatory response is based on their immunomodulating effects. A recent phase II trial showed that in patients with severe community-acquired pneumonia requiring mechanical ventilation and with a high inflammatory pattern, the adjunctive therapy with a new intravenous immunoglobulin preparation containing 18% of IgM reduced the mortality by about 20% compared to the placebo [55]. Two phase III trials in community-acquired pneumonia and COVID-19 patients are underway to confirm the results observed with the use of this new preparation. In septic patients with hyperinflammation or toxic shock, the timing for IVIg therapy may have a substantial role, and two large studies indicated that earlier therapy (within 12 h) may decrease the mortality risk [52]. 

Scenario 2, late phase (see Figure 2): COVID-19 interstitial pneumonia and suspected immune dysfunction/immune paralysis.

Description of the scenario: Patient requiring mechanical ventilation for progressive worsening of acute respiratory failure several days (7–10) after the occurrence of COVID-19 interstitial pneumonia and with low plasma levels of such inflammatory parameters as C-reactive protein, ferritin and IL-6 and persistent lymphopenia. 

Questions: (1)In COVID-19 with progressive worsening of respiratory failure, suspected immune dysfunction/immune paralysis and low plasma levels of immunoglobulins, how appropriate is the replacement therapy with polyclonal intravenous immunoglobulins to prevent secondary infections?Consensus rating: appropriate; median score, 7 (IQR, 6–8); disagreement: no.(2)In COVID-19 with progressive worsening of respiratory failure, suspected immune dysfunction/immune paralysis and with the decision to use polyclonal intravenous immunoglobulins, how appropriate is the use of preparations including also the IgM component?Consensus rating: appropriate; median score, 8 (IQR, 7–8); disagreement: no.

Rationale:

In COVID-19 patients, the anti-inflammatory response mediated by molecules such as IL-10, IL-4 and TGF-β is finalized to balance the initial proinflammatory response described in several models. Therefore, as previously described, in COVID-19 patients, several alterations in innate and adaptive immunity occur, including marked lymphopenia. However, dysregulated and/or persistent activation of the anti-inflammatory components, often with the addition of anti-inflammatory treatments (e.g., steroids, cytokine-blocking agents and others), may cause a severe failure of the immune system defined in sepsis models as immune paralysis and characterized by lymphopenia with marked T cell exhaustion, alteration of cytokines profile, inadequacy of antigen-presenting mechanisms and dysfunction and apoptosis of B and T lymphocytes [15,20]. Patients with immune paralysis are unable to mount an appropriate inflammatory response and become prone to viral reactivation and secondary or breakthrough infections, mostly by opportunistic agents. A high rate of secondary bacterial and viral infections has been reported by numerous studies in COVID-19 patients, especially in those requiring mechanical ventilation and ICU admission [56,57].

In patients with sepsis, low plasma levels of Ig have been frequently reported and are closely related to the severity of the underlying conditions and poor outcomes [58]. Moreover, the kinetics of plasma IgM in the first days after sepsis is different in survivors and non-survivors of septic shock [30]. In an observational study involving 62 COVID-19 patients with ARDS admitted to ICU [59], the authors observed that in patients with IgG levels below 7 g/L, the 90-day mortality was higher than in patients whose levels exceeded 7 g/L.

In immunocompromised patients, IVIg therapy is commonly used for preventing infections and sepsis [26,32]. In patients with hypogammaglobulinemia after heart transplantation, prophylactic IgG therapy decreased the incidence of severe secondary infections, but the same was not observed after lung transplantation [60,61]. Indeed, a more recent meta-analysis showed that the prophylactic use of the IVIg therapy was associated with a better survival rate in heart and lung recipients with hypogammaglobulinemia [62].

Scenario 3, sepsis by bacterial superinfection: COVID-19 interstitial pneumonia with septic shock by hospital-acquired superinfection. 

Description of the scenario: Patient with acute respiratory failure by COVID-19 interstitial pneumonia and occurrence of septic shock sustained due to a nosocomial bacterial infection.

Questions: (1)In COVID-19 patients with septic shock due to nosocomial acquired infections, how appropriate is the adjunctive therapy with polyclonal intravenous immunoglobulins?Consensus rating: appropriate; median score, 7 (IQR, 6–8); disagreement: yes.(2)In COVID-19 patients with septic shock due to nosocomial acquired infections with the decision to use adjunctive therapy with polyclonal intravenous immunoglobulins, how appropriate is the use of preparations including the IgM component?Consensus rating: appropriate; median score, 9 (IQR, 8–9); disagreement: no.

Rationale:

Critically ill COVID-19 patients are prone to secondary infections because of many factors such as invasive mechanical ventilation, long ICU stay, anti-inflammatory therapies, SARS-CoV-2-induced immune suppression. Worldwide studies have reported a high rate of ventilator-associated pneumonia ranging from 23% to 37% and of secondary bacteremia in ICU-admitted patients. In addition, as for other categories of complicated ICU patients, infections caused by multidrug-resistant bacteria or *Aspergillus* spp. are frequent in COVID-19 patients [56,63,64,65]. Therefore, sepsis and septic shock often complicate the ICU stay and are a major cause of mortality [66,67,68]. Despite the lack of definitive evidence, IVIg adjunctive therapy has been used for 30 years, and meta-analysis indicates a possible benefit in septic patients, with better results when using preparations enriched with the IgM component [51,54]. Many authors support adjunctive therapy in specific phenotypes of septic patients such as those with hyperinflammation, overwhelming shock or blunted inflammatory response [69]. Indeed, most clinical experiences refer to the former phenotypes while the data in immunocompromised septic patients are scarce and not definitive. The use of IgM preparations has also been demonstrated to be effective in ICU patients with MDR infections, particularly the ones caused by Gram-negative microorganisms [30,51]. 

## 5. Conclusions

The alterations in immune and inflammatory responses observed in the different phases of COVID-19 are similar to those observed in septic patients. The key role of endogenous immunoglobulins in host response and the robust experience in immunocompromised and septic patients make adjunctive therapy with IVIg attractive in COVID-19, particularly in hospitalized patients with severe respiratory failure. Despite the paucity of the existing data, the structured consensus identified rationale for the use of IVIg therapy in these patients with immune paralysis for preventing secondary infections and in patients with septic shock caused by nosocomial infections. Several appropriate studies are underway (ClinicalTrials.gov identifiers NCT04432324, NCT04500067, NCT04576728, NCT04350580, NCT04350580) for defining the patients who can benefit the most and the appropriate time and dose of IVIg therapy.

## Figures and Tables

**Figure 1 jcm-10-03500-f001:**
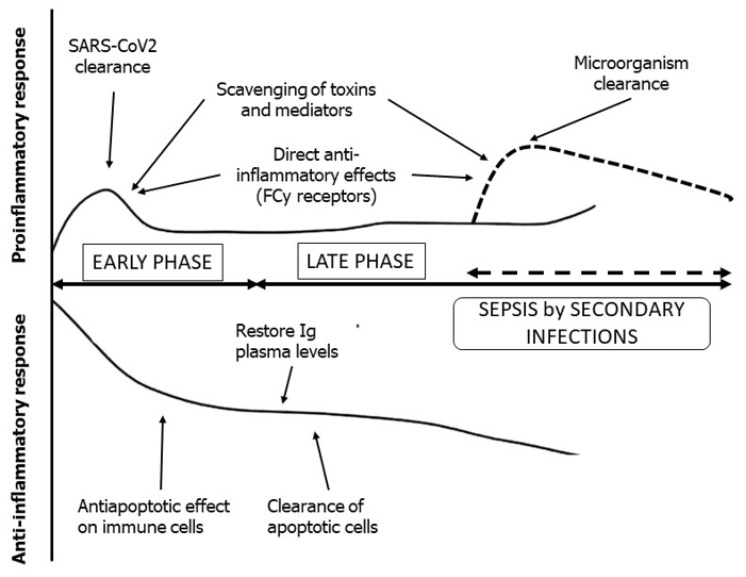
Host immune–inflammatory response in COVID-19 (modified from [21]). The potential role of immunoglobulins in pro- and anti-inflammatory response and the three scenarios (i.e., early, late, sepsis by secondary infections) identified by the panel are also reported.

**Figure 2 jcm-10-03500-f002:**
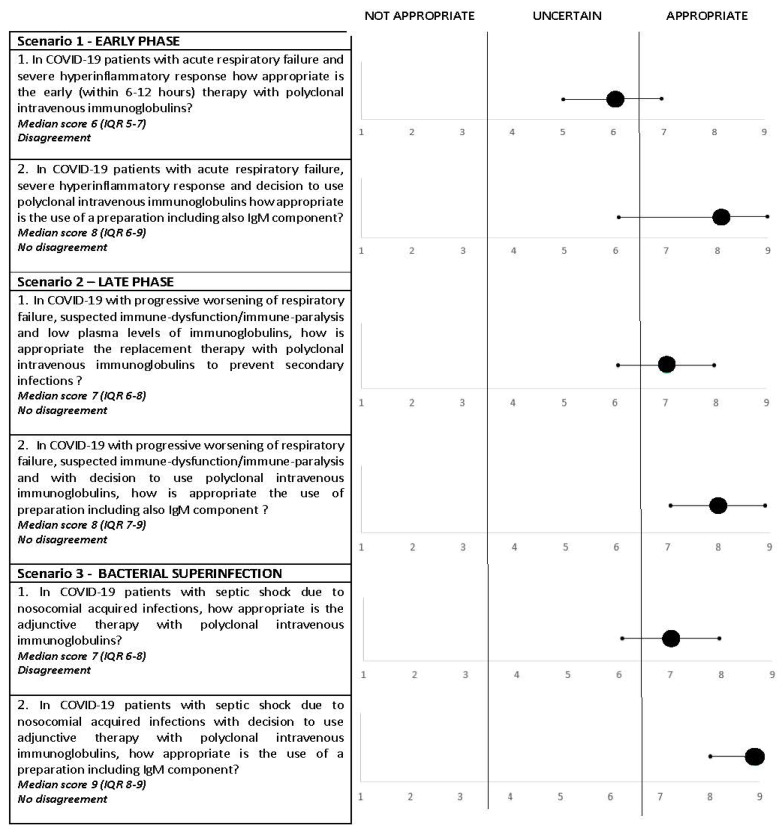
Results of the appropriateness evaluation of polyclonal intravenous immunoglobulin adjunctive therapy in the three scenarios defined by the panelists.

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
