# Peer review of "Rationale for Polyclonal Intravenous Immunoglobulin Adjunctive Therapy in COVID-19 Patients: Report of a Structured Multidisciplinary Consensus"

_jcm, 2021, doi:10.3390/jcm10163500_

Round 1

Reviewer 1 Report

Authors present an interesting and thoughtful paper, regarding a controversial issue in the treatment of COVID patients of 3 different phenotypes. An extensive review of the state of the art has been carried out that supports  a adjunctive role of Igs in their treatment. The discussion of such an issue is very welcome at this moment, given the controversial data that has been generated from COVID patients within the last 12 months including those obtained from normoimmune as well as immunodeficient patients.

Author Response

We really thank the reviewer for the comment.

Reviewer 2 Report

This manuscript reported the consensus of a panel of Corona expert on the use and therapeutic effect of IvIg in adjuvant setting. The literature cited is fitting in almost all aspects of the clinical phase of the disease supporting the conclusion. However, the mechanism of IvIg immunoregulation is only touched in three citations (26, 28 and 48). For example, recent proposed pathways if IvIg role in immune-modulation are missing. [1; 2; 3; 4] [1] P. Tabarsi, S. Barati, H. Jamaati, S. Haseli, M. Marjani, A. Moniri, Z. Abtahian, A. Dastan, S. Yousefian, R. Eskandari, A. Saffaei, F. Monjazebi, A. Vahedi, and F. Dastan, Evaluating the effects of Intravenous Immunoglobulin (IVIg) on the management of severe COVID-19 cases: A randomized controlled trial. International immunopharmacology 90 (2021) 107205. [2] A. Yaqinuddin, A.R. Ambia, T.A. Elgazzar, M.B.M. AlSaud, and J. Kashir, Application of intravenous immunoglobulin (IVIG) to modulate inflammation in critical COVID-19 - A theoretical perspective. Med Hypotheses 151 (2021) 110592. [3] A. Nguyen, Y. Repesse, M. Ebbo, Y. Allenbach, O. Benveniste, J.M. Vallat, L. Magy, S. Deshayes, G. Maigne, H. de Boysson, A. Karnam, S. Delignat, S. Lacroix-Desmazes, J. Bayry, and A. Aouba, IVIg increases interleukin-11 levels, which in turn contribute to increased platelets, VWF and FVIII in mice and humans. Clin Exp Immunol 204 (2021) 258-266. [4] H. Kohler, and S. Kaveri, How IvIg Can Mitigate Covid-19 Disease: A Symmetrical Immune Network Model. Monoclon Antib Immunodiagn Immunother 40 (2021) 17-20.

Author Response

RESPONSE: We thank the reviewer for the useful suggestions that we included in the manuscript (see Section 2. Immunoglobulin Adjunctive Therapy, Page 3).

Reviewer 3 Report

This is provocative and timely discussion of the experience with the use of IVIG to treat COVID 19 patients in ICU setting. The use of polyclonal IVIG preparations, both purified IgG and IgM enriched has been a controversial topic for many years with many conflicting outcomes based on direct studies as well as meta-analysis (Crit Care Med 2007 Dec;35(12):2855-6) (Crit Care Med 2007 Dec;35(12):2852). Unfortunately, this paper does little to clarify the controversies.

Specific comments:

  1. The authors should clearly point out the variability in the study designs in the published studies in patients with COVID 19. In some cases the IVIG was given at immune modulatory doses and in others immune prophylaxis with varying timing of administration. Furthermore, many included the use of corticosteroids therefore the interpretation of the results are difficult.
  2. There is little mention of the adverse reactions or cost associated with IVIG infusions, particularly as it relates to IgM enriched products.
  3. It is not clear what material was available to the panelist that formed the basis of their recommendations.
  4. The paper should clearly differentiate studies done with IgG based versus IgM enriched products in the text.
  5. The conflict of interest section should point out that Biotest, the sponsor of the meeting manufactures and markets IgM enriched IVIG

Author Response

This is provocative and timely discussion of the experience with the use of IVIG to treat COVID 19 patients in ICU setting. The use of polyclonal IVIG preparations, both purified IgG and IgM enriched has been a controversial topic for many years with many conflicting outcomes based on direct studies as well as meta-analysis (Crit Care Med 2007 Dec;35(12):2855-6) (Crit Care Med 2007 Dec;35(12):2852). Unfortunately, this paper does little to clarify the controversies.

RESPONSE: We are well aware, as reported in the text, that the use of IVIg in critically ill patients (especially in sepsis) is a matter of debate for many years, but it was beyond the aim of the consensus (and manuscript) to enter in this debate.

Specific comments:

  1. The authors should clearly point out the variability in the study designs in the published studies in patients with COVID 19. In some cases the IVIG was given at immune modulatory doses and in others immune prophylaxis with varying timing of administration. Furthermore, many included the use of corticosteroids therefore the interpretation of the results are difficult.

RESPONSE: We fully agree with these considerations. A paragraph including these points has been inserted in the text (see Section 2 Immunoglobulin Adjunctive Therapy, at the bottom of Page 3)

  1. There is little mention of the adverse reactions or cost associated with IVIG infusions, particularly as it relates to IgM enriched products.

RESPONSE: We added a focus on adverse reactions in section 2 “Immunoglobulin Adjunctive Therapy” Page 3.

  1. It is not clear what material was available to the panelist that formed the basis of their recommendations.

RESPONSE: We performed a systematic review of the literature and extrapolated a synthesis of evidence made available for all the panellists before the meeting. During the meeting, the synthesis of evidence was presented and discussed with the panellists. We better explained this in the methods section (page 4)

  1. The paper should clearly differentiate studies done with IgG based versus IgM enriched products in the text.

RESPONSE: We better specified in the text this issue as suggested (see Section 2 Immunoglobulin Adjunctive Therapy, Page 3)  

  1. The conflict of interest section should point out that Biotest, the sponsor of the meeting manufactures and markets IgM enriched IVIG

RESPONSE: Added in the “conflict of interest” section as requested (see page 8)

Round 2

Reviewer 3 Report

revisions are acceptable